# Exploring the feasibility of a network of organizations for pain rehabilitation: What are the lessons learned?

**Cynthia Lamper**[1]*, **Ivan P. J. Huijnen**[1,2]*, **Mariëlle E. A. L. Kroese**[3], **Albère J. Köke**[1,2], **Gijs Brouwer**[3], **Dirk Ruwaard**[3], **Jeanine A. M. C. F. Verbunt**[1,2]

**1** Faculty of Health, Medicine and Life Sciences, Department of Rehabilitation Medicine, Functioning, Participation & Rehabilitation, School for Public Health and Primary Care (CAPHRI), Maastricht University, Maastricht, The Netherlands, **2** Centre of Expertise in Rehabilitation and Audiology, Adelante, Hoensbroek, The Netherlands, **3** Faculty of Health, Medicine and Life Sciences, Department of Health Services Research, School for Public Health and Primary Care (CAPHRI), Maastricht University, Maastricht, The Netherlands

* cynthia.lamper@maastrichtuniversity.nl (CL); ivan.huijnen@maastrichtuniversity.nl (IPJH)

## Abstract

### Background and aims

Integration of care is lacking for chronic musculoskeletal pain patients. Network Pain Rehabilitation Limburg, a transmural health care network, has been designed to provide integrated rehabilitation care from a biopsychosocial perspective to improve patients' levels of functioning. This feasibility study aims to provide insight into barriers and facilitators for the development, implementation, and transferability.

### Methods

This study was conducted with a three-phase iterative and incremental design from October 2017 to October 2018. The network comprises two rehabilitation practices, and three local primary care networks, with a general practitioner together with, a mental health practice nurse, and a physiotherapist or exercise therapist. These stakeholders with a random sample of participating patients took part in evaluations, consisting of interviews, focus groups, and observations. Field notes and observations were recorded during meetings. The Consolidated Framework for Implementation Research guided data collection and analysis. Results were used to refine the next phase.

### Results

According to health care professionals, guidelines and treatment protocols facilitate consistency and transparency in collaboration, biopsychosocial language, and treatment. One mentioned barrier is the stigmatization of chronic pain by the general population. In regular care, approaches are often more biomedical than biopsychosocial, causing patients to resist participating. The current organization of health care acts as a barrier, complicating implementation between and within practices. Health care professionals were enthusiastic about

**Data Availability Statement:** Most relevant data are within the paper and Supporting information files. The other data for this study consist of transcripts of focus groups and interviews with participants and contain identifying items, and are

therefore sensitive to privacy issues. As participants only allowed the interviews under promise of anonymity, we are expressly forbidden by the participants to make the full content of the focus groups and interviews public. Anonymized excerpts from the full transcripts can be made available to qualified researchers by request to the medical ethical committee of Zuyderland, who can be contacted at metc@zuyderland.nl.

**Funding:** Funding was received from the non-profit private health insurers: CZ (in Dutch: Centraal Ziekenfonds; https://www.cz.nl), VGZ (in Dutch: Stichting Volksgezondheidszorg Zuid Nederland; https://www.vgz.nl/), Zilveren Kruis (https://www.zilverenkruis.nl). This applies for: Ivan P.J. Huijnen Albère J. Köke, Jeanine A.M.C.F. Verbunt.

**Competing interests:** IH, AK, and JV report grants from Health Insurance Companies CZ, VGZ and Achmea, during the conduct of the study. The other authors declare that there is no conflict of interest. This does not alter our adherence to PLOS ONE policies on sharing data and materials.

the iterative, bottom-up development. A critical mass of participating organizations is needed for proper implementation.

## Conclusion

Network Pain Rehabilitation Limburg is feasible in daily practice if barriers are overcome and facilitators of development, implementation, and transferability are promoted. These findings will be used to refine Network Pain Rehabilitation Limburg. A large-scale process and effect evaluation will be performed. Our implementation strategies and results may assist other health care organizations aspiring to implement a transmural network using a similar model.

## Trail registration

**Registration number**: NTR6654 or https://trialsearch.who.int/Trial2.aspx?TrialID=NTR6654.

## Introduction

Nineteen percent of adults in Europe have moderate to severe chronic pain [1]. The most widely reported complaint is chronic musculoskeletal pain (CMP), representing a complex interaction of biopsychosocial components, varying in complexity between patients [2,3]. CMP can have a significant impact on patients' daily activities and, therefore, rehabilitative treatments are needed [4]. Most patients with CMP have had this pain for more than two years [5]. Due to a high burden of disease and work absence in these patients, the direct and indirect costs of CMP are estimated at 20 billion euros in the Netherlands yearly [6–9]. While 60–74% of Dutch CMP patients receive treatment, 34–79% of these patients feel that this is inadequate [1,4,10,11]. Such patients continue seeking a solution for their CMP, resulting in high medical resource consumption by this group [12].

A possible explanation for the level of resource consumption is that the complexity of the patient's pain problem does often not match the treatment delivered, resulting in over- or under-treatment [13]. This mismatch can be explained by three factors. Firstly, understanding of biopsychosocial treatment of CMP varies amongst health care professionals (HCPs), decision-makers, and the public. Secondly, clinical decision-making, classification of complexity, and referrals are based on medical history and clinical experience, with huge inter-physician variation. Although earlier studies have shown inter-rater reliability in classifying the complexity of pain problems by rehabilitation physicians (RPs) to be at least questionable, the use of objective measures by general practitioners (GPs) and RPs to diagnose and classify patients with complex problems is scarce [14–17]. Thirdly, treatment approaches, including dosage and content, delivered through all types of care providers, are often not adequately tailored to the level of complexity of the pain problem [18]. Therefore, patients with CMP often do not receive the right care, at the right place, at the right time, as described in the National Care Standard for Chronic Pain, the Netherlands [6].

To overcome this problem, integrated transmural health care networks, including all health care settings, might have a beneficial role [19]. Integrated transmural care is what is described by the World Health Organization as "*the management and delivery of health services so that clients receive a continuum of preventive and curative services, according to their needs over time*

*and across different levels of the health system*"[20]. It is most often directed towards bridging the gap between care providers in different levels of care, for example between primary and secondary care. The World Health Organization recommends networks in integrated transmural rehabilitation care as future developments [21].

The transmural Network Pain Rehabilitation Limburg (NPRL) was designed to implement rehabilitation care according to the National Care Standard for Chronic Pain in the province of Limburg in the Netherlands [6]. To overcome mismatches in current CMP rehabilitation care, NPRL has an unambiguous view: integrated matched care, biopsychosocial treatment protocols for primary, secondary and tertiary care, guidelines for referral and coordination, and a continuous focus on improvement of care. Matched care comprises identifying patients at higher risk. However, unlike stratified care, it tailors the intervention to the individual patient's specific existing complaints and risk [22,23]. In NPRL, patients can be referred by their GP, a primary care physical therapist or the RP. In the Netherlands, patients can also visit a primary care therapist (e.g. physiotherapist or occupational therapist) directly without a referral. If patients enter the health care system in this way, therapists can screen the patient as to their suitability for treatment within NPRL. Treatment is offered, based on complexity profiles, by primary, secondary or tertiary care members of the NPRL. In order to develop and implement NPRL in daily care, a feasibility study was performed. Details of the protocol of this study are described elsewhere [24].

The aim of the feasibility study was to provide insight into barriers and facilitators for the development, implementation, and transferability of NPRL, and to provide insight into its perceived value and acceptability.

## Methods

### Study design

This feasibility study had an iterative and incremental design based on key principles of user-centred design [25]. It was conducted from October 2017 to October 2018 in the South-East region of the province of Limburg in the Netherlands. As NPRL is a complex intervention, the UK Medical Research Council (MRC) Framework was used as guidance for development (Phase 1), implementation (Phase 2), and transferability (Phase 3) of NPRL [26]. The barriers and facilitators that emerged during the evaluations of previous phases were used to refine elements of NPRL in the next phase. HCPs and patients actively participated in evaluations, leading to adjustments in daily health care practice.

Topic lists for individual interviews and focus group sessions were constructed, and the results were analysed deductively, in accordance with the Consolidated Framework for Implementation Research (CFIR), following a directed content analysis method [27]. This framework, published by Damschroder et al. (2009), is an overarching list of constructs to verify the design's efficacy across multiple contexts when implementing a complex multi-component intervention such as NPRL, with rapid-cycle evaluation [27,28]. It consists of five major domains (intervention characteristics, outer setting, inner setting, characteristics of individuals, and process of implementation) with 39 underlying constructs and sub-constructs that can potentially influence implementation efforts. The study was reported using the COnsolidated criteria for REporting Qualitative research items (COREQ) [29].

### Ethics and dissemination

Written and verbal informed consent was obtained from all patients and HCPs before the start of the interview or focus group. The verbal informed consent was recorded. Ethical approval for this study was granted by the Medical Ethics Committee Z, the Netherlands, METC 17-N-

133. Dissemination includes publications and presentations at regional, national and international conferences.

## Organization of care in Network Pain Rehabilitation Limburg

NPRL is described more extensively in Lamper et al. (2019) (a summery can be found in S1 File): it was developed by a project team consisting of the authors CL, IH, AK, JV and an advisory board consisting of interested HCPs [24]. The project team as well as advisory board consisted of HCPs of different disciplines and they had experience in the development of treatment protocols. The main aim of NPRL was to provide integrated, biopsychosocial rehabilitation care for the right patients with CMP at the right place and at the right time [22]. The content is based on the National Care Standard for Chronic Pain, which proposes a matched care approach in an integrated transmural network for patients with CMP [6]. Based on the matched care approach, HCPs from different disciplines participated and provided several treatments. For a detailed overview of the structure and organization of the health care system in NPRL see Fig 1 published in Lamper et al. (2019) [24]. Elements were integrated into NPRL to reach the overall goal. Two assessment tools supported the decision-making for problem and complexity mapping and treatment selection, based on the patient's biopsychosocial profile. GPs and therapists in primary care used Assessment Tool 1 (S2 File); Assessment Tool 2 was used by RPs in secondary and tertiary care. In the individualized treatment plan, the patient together with the HCP set activity- and participation-related goals. An e-health application was integrated into matched care protocols for every setting with the primary goal of supporting pain education and self-management by the patient.

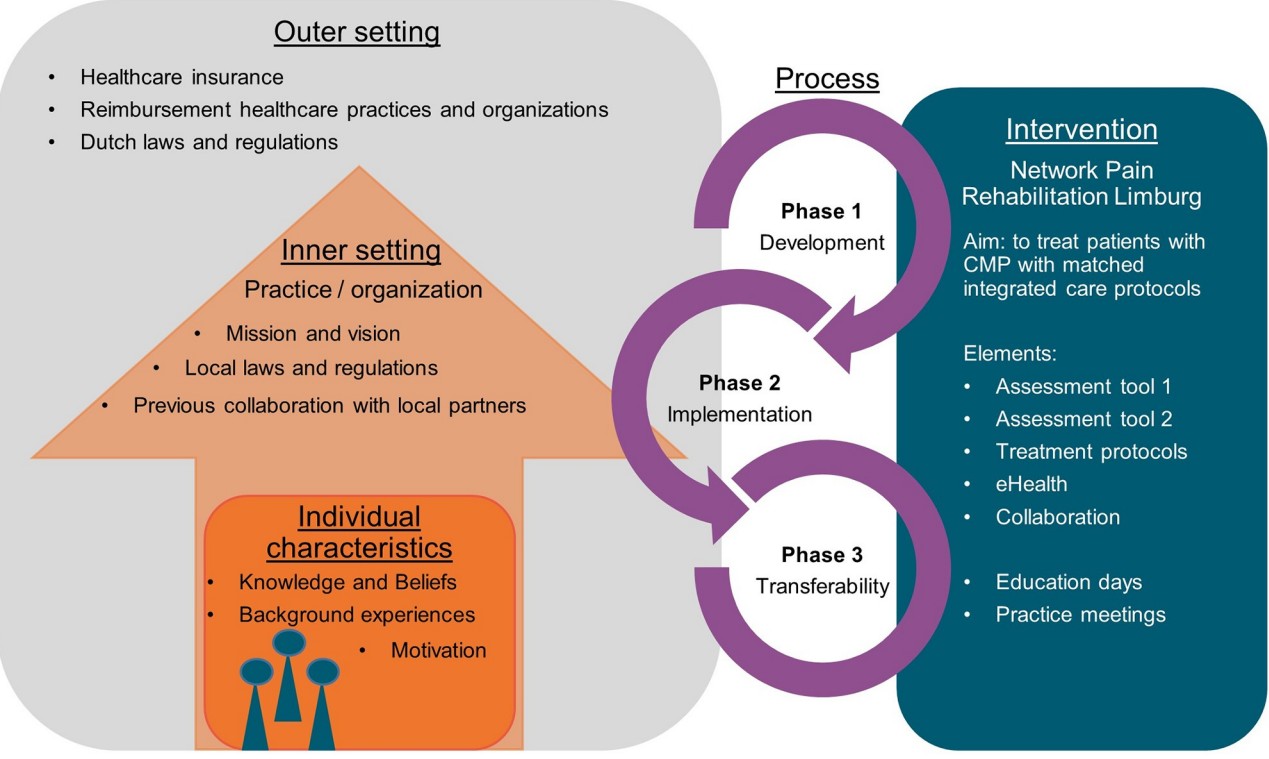

**Fig 1. Consolidated framework for integrated care adjusted for Network Pain Rehabilitation Limburg.**

## Health care professionals

In this transmural NPRL, HCPs from all health care levels with a prior interest in CMP were included as participants. In primary care, local networks were set up in villages or city districts with local HCPs. For inclusion in a local network, it was necessary to have a GP (general practitioner, family doctor) participating, with in addition at least one physiotherapist (PT) or exercise therapist (ET), and, optionally, a mental health practice nurse (MHPN). Initially, six local networks were contacted for participation of which three were included in this study (1: one GP, two PTs; 2: one GP, one MHPN, one PT, one ET; 3: one GP, two PTs). One local network stopped participating because of lack of time, but their PT participated in focus groups. The other two local networks were not included because they did not meet the inclusion criteria. GPs were excluded if they participated in fewer than two out of three education days; therapists were excluded if they participated in fewer than three out of four education days.

In addition, a private outpatient rehabilitation clinic (two RPs; one psychologist (PSY)) and a specialized rehabilitation clinic in tertiary care (two RPs; one physician assistant (PA); one nurse practitioner (NP); one treatment coach (TC)) participated in this study. One outpatient rehabilitation clinic did not meet the inclusion criteria, but its RP participated in this study. HCPs were educated in the clinical guidelines as described in the National Care Standard for Chronic Pain and in the study process before its start and participated in focus groups at the end of each phase. Moreover, every six to eight weeks, all HCPs within one local network in primary care received supervision at their own practice. Secondary and tertiary care organizations had to meet the criteria of the Position Paper 'Medical Specialist Rehabilitation for chronic musculoskeletal pain' (2017) [30]. Practices and organizations were excluded when they were unable to implement the different elements of NPRL.

## Patients

Of the 58 patients participating in NPRL, nine registered in primary care were randomly asked by telephone to participate in a focus group, with six agreeing. They had to be older than 18 years at the start of the study, have musculoskeletal pain that was (expected to become) chronic as indicated by a HCP, and be treated by participating HCPs who also participated in a focus group. Their treatment aim had to be improvement of daily functioning despite pain. They were excluded from the study if there was any suspicion of a biomedical (orthopaedic, rheumatic, or neurological) disease that could explain the current pain complaints and could be treated by adequate existing therapy. In addition, they were excluded if there was any (underlying) psychiatric disease (Personality disorder, schizophrenia, or clinical depression) that limit the possibility for behavioral change. Pregnancy and inadequate Dutch literacy were also exclusion criteria.

## Data collection

To achieve triangulation, data were collected with different methods, such as field notes or observations during meetings, individual interviews, focus groups, and questionnaires, which were all combined to get more insight into barriers and facilitators (S3 File) [31]. The observations and individual interviews were conducted by CL. The focus groups were all led by GB, an independent researcher not aligned with the project; CL, the main researcher of the project, was the observer and made field notes during and after the focus groups. At the time of data collection, CL and GB were PhD students and had 1–2 years of experience in health care sciences. Both the individual interviews and focus groups, for which semi-structured guides were made based on CFIR, were audio-recorded and transcribed verbatim. All interviews took

place at the work place of the HCP; all focus groups were organized in the tertiary care organization.

**Phase 1: Development of NPRL.**   Phase 1 was conducted from October 2017 to February 2018. The goal of the phase was to design and develop the content of NPRL and to educate participating HCPs. In the evaluation, the focus was on the perceived barriers and facilitators of this development process. Focus groups were held with HCPs from the local networks, and interviews were held with HCPs from secondary and tertiary care organizations at the end of the phase to gather information about their experiences with the informative meetings and education days, and about their expectations and current experiences of working in NPRL.

**Phase 2: Implementation of NPRL.**   Phase 2 was conducted from February to June 2018. The goal during this phase was to specify the content and to implement NPRL in daily practice. During the evaluations, barriers and facilitators regarding the implementation of NPRL were identified. Focus groups were held with HCPs from the local networks and secondary and tertiary care practices combined, and the individual interviews with the MHPN and a RP. The focus was on their current experiences of working in NPRL, and implications and recommendations for the implementation strategy in the practices.

**Phase 3: Transferability of NPRL.**   Phase 3 was conducted from June to October 2018. The goal of this phase was to organize care in daily practice. Evaluations focused on barriers and facilitators for the transferability of NPRL beyond the pilot region. At the end of the phase, focus groups and interviews were organized with the HCPs to collect more information on current experiences of working with NPRL, and implications and recommendations for the implementation strategy regarding the transferability of NPRL to practices. Additionally, information was gathered about satisfaction with NPRL and its effect on work life. Moreover, a focus group with six patients was organized to develop more insight into the perceived quality of care, their experiences with NPRL, and barriers and facilitators associated with different elements of NPRL observed by them.

**Overall.**   After completing treatments, HCPs submitted predefined questionnaires or logbooks about the treatment (number of consultations, barriers and facilitators during treatment, achievement of treatment goal) of each individual patient. In addition to the information collected in the three phases, CL kept a logbook of barriers and facilitators of NPRL mentioned by participants, the field notes and observations in these logbooks being the result of discussions with different HCPs, patients, and researchers.

### Data analysis

A content analysis with mostly a deductive approach was used, with the CFIR as coding framework [27,32]. After familiarization with the data, definitions for the CFIR constructs based on NPRL were compiled (Fig 1) and used to guide data analysis. For each construct, CL composed codes based on the data, using NVivo software (NVivo version 11.1.0.411). After analyzing all data, the codes were summarized in barriers and facilitators per construct. GB performed a peer review of the analysis by verifying 20% of the interviews and focus groups. When disagreement occurred, the research team was consulted. The coding process was guided by consensual qualitative research methods [33,34]. Moreover, two HCPs performed a cross-check for interim findings by providing feedback on the results.

### Results

Table 1 displays the content, duration, and disciplines involved in each interview and focus group. Five focus groups and six interviews with 21 HCPs from different disciplines, and one

**Table 1. Description of the content, duration, and discipline involved for the focus groups and interviews.**

| | | | | Phase 1 | | | | | | Phase 2 | | | | Phase 3 | |
|---|---|---|---|---|---|---|---|---|---|---|---|---|---|---|---|
| | | | | FG 1 | FG 2 | INT 1 | INT2 | INT 3 | INT 4 | FG 3 | FG 4 | INT 5 | INT 6 | FG 5 | FG 6 |
| Duration | | | | 1h45m | 1h26m | 37m | 36m | 29m | 28 m | 1h35m | 1h24m | 27m | 49m | 1h34m | 1h48m |
| Goal | | | | | | | | | | | | | | | |
| Experiences with the organization of rehabilitation care for patients with CMP before participating in NPRL | | | | | | x | x | x | x | | x | x | x | | |
| Expectations for participation in NPRL | | | | x | x | x | x | x | x | | | | x | | |
| Barriers and facilitators of the development process | | | | x | x | | | | | | x | x | | | |
| Barriers and facilitators of the implementation strategy | | | | | | | | | | x | | x | | x | |
| Expected barriers and facilitators of the transferability phase | | | | | | | | | | | | | | x | |
| Current experiences being a patient in NPRL (eg. eHealth, healthcare professional skills, referral, treatment, feeling of collaboration) | | | | | | | | | | | | | | | x |
| | Discipline | Gender | Exp. (yrs) | | | | | | | | | | | | |
| P1 | PT | F | 2.5 | x | | | | | | | x | | | | |
| P2 | PT | M | 0.5 | x | | | | | | | | | | x | |
| P3 | PT | M | 34 | x | | | | | | | x | | | | |
| P4 | PT | M | 38 | x | | | | | | | | | | | |
| P5 | PT | F | 7 | | x | | | | | | x | | | | |
| P6 | PT | M | 30 | | x | | | | | | | | | | |
| P7 | PT | M | 33 | | x | | | | | | | | | | |
| P8 | ET | F | 25 | | x | | | | | | x | | | | |
| P9 | PNMH | F | - | | | | | | | | | x | | | |
| P10 | GP | M | 10 | x | | | | | | | x | | | | |
| P11 | GP | M | 31 | x | | | | | | | | | | | |
| P12 | GP | M | 8 | | x | | | | | | | | | x | |
| P13 | PSY-2 | F | - | | | | | x | | | | | | | |
| P14 | RP-2 | F | 6 | | | | | | x | | x | | | | |
| P15 | RP-2 | M | - | | | | | | | | x | | | x | |
| P16 | RP-3 | F | - | | | x | | | | | | | | | |
| P17 | RP-3 | F | <1 | | | | x | | | | | | | | |
| P18 | PA-3 | M | 15 | | | | | | | | x | | | x | |
| P19 | NP-3 | F | 6 | | | | | | | | x | | | x | |
| P20 | TC-3 | F | - | | | | | | | | | | | x | |
| P21 | RP-2 | M | - | | | | | | | | | | x | | |
| P30 | PNT | F | n.a | | | | | | | | | | | | x |
| P31 | PNT | F | n.a | | | | | | | | | | | | x |
| P32 | PNT | F | n.a | | | | | | | | | | | | x |
| P33 | PNT | M | n.a | | | | | | | | | | | | x |
| P34 | PNT | F | n.a | | | | | | | | | | | | x |
| P35 | PNT | M | n.a | | | | | | | | | | | | x |

FG: Focus group; INT: Interview; EXP: Years experience; PT: Physiotherapist; ET: Exercise therapist; PNMH; practice nurse mental health; GP: General practitioner; PSY-2: Psychologist secondary care; RP-2: Rehabilitation physician secondary care; RP-3: Rehabilitation physician tertiary care; PA-3: Physician assistant tertiary care; NP-3: Nurse practitioner tertiary care; TC-3: Treatment coach tertiary care; PNT: Patient; F: Female; M: Male; -: Unknown; n.a.: Not applicable.

focus group with six patients were held. The results were analysed and described, based on the domains of the CFIR (Fig 1).

## Intervention

**Assessment Tool 1 & Assessment Tool 2.** HCPs in primary care found Assessment Tool 1 too time-consuming, because of the extra burden for the patient and the extra time for the consultation itself. A further consultation to discuss the results with patients was not desirable. *"I have no time to discuss the results with the patient in an extra consultation." (P12: GP, FG2).* In addition, several HCPs thought that the results of Assessment Tool 1 were not in line with their conclusions, based on their observations, experience, and assessments of patients. All secondary and tertiary rehabilitation physicians reported that Assessment Tool 2 supported their knowledge and assessments, but found its administration too time-consuming.

**Treatment protocols and guidelines.** According to most of the HCPs, the treatment protocols and guidelines within NPRL provide a common biopsychosocial language and transparency in treatment duration, intensity, and content: *"In my opinion, in NPRL the treatment approach is more explicit and I know these are the steps to take to achieve a result compared to usual care." (P2: PT, focus group 5).* HCPs in local networks indicate that the protocols and guidelines provide a clear overview of the total approach in CMP management. Patients are more familiar and better informed about the content of various treatments in transmural care, compared with those treated before NPRL started. Due to the restricted number of consultations prescribed in the treatment protocol of NPRL as compared to care as usual, some therapists in primary care indicated fear that this would lead to a drop in income from that achieved before NPRL's start.

HCPs have different personal preferences and opinions about the freedom in implementation of the treatment protocol and guidelines. Some HCPs felt that this freedom was desirable as it could be adjusted to the local organization of the primary care practices: *"Currently, it is not a tight protocol, of which we are the executors, searching for patients who fit. And I think the strength lies in the fact that we as HCPs can decide how to implement the knowledge that we have gained in the area of chronic pain, in a way that will fit into our daily care routines. That is an essential difference, as P10 [GP] said" (P3: PT, FG1)).* This freedom might be an important facilitator, according to the HCPs, if NPRL is to be implemented in the Netherlands. Other HCPs underlined the importance of standardization, with fixed treatment protocols, as they wanted more control of the treatment of this complex patient population.

**eHealth application.** The participants indicated that e-health has a central position in NPRL: it facilitates and supports the patient in the treatment process, and collects biopsychosocial information about the patient. According to primary HCPs, the eHealth application is user-friendly and the collected information derived from assessment reduces the duration of consultations. However, some GPs see the collection of extra information as an extra burden for patients. Other barriers of the eHealth application mentioned were: the slow speed of the two-step authentication log-in facility, lack of an overview of the steps in the treatment, and difficulties in using the chat function in daily practice because no HCP is assigned to keep track of it. These barriers meant that some HCPs had little experience of using the eHealth application. RPs saw no added value of the diary function in the eHealth application as they did not see patients frequently enough during rehabilitation to integrate it into treatment.

All patients agreed that the eHealth application (existing of pain education and self-management exercises) stimulated them to adhere to the treatment. Both the graphs of their daily activity and the education material provided were especially motivating: *"The most important advantage of the eHealth application is the diaries: they keep me motivated. I like the competition*

*with myself to be more active" (SP35, patient, FG6)*. On the other hand, it was hard for some patients to complete the diary daily so they missed the added value of this daily returning questionnaire. Also, some patients could not participate in this study because they had no internet connection.

**Collaboration.** Some therapists appreciated the fact that interdisciplinary collaboration with GPs and MHPNs working closely together is a major pillar of NPRL. However, collaboration with GPs was perceived as difficult as it takes a lot of effort to contact them for consultation and discussion. At the end of Phase 1, some HCPs reported no change in levels of collaboration in local networks of NPRL. In Phases 2 and 3, more change in collaboration was reported, though this was still not optimal. Better interdisciplinary collaboration was achieved in local networks based in one site, compared to those in which the GP and MHPN were located at a different address from the therapists. According to the HCPs in primary care, interdisciplinary collaboration in a local network will facilitate treatment of patients with more complex pain complaints, leading to a decrease in referrals to secondary or tertiary care. Also, they felt that young and dynamic teams would facilitate implementation. In the future, it is hypothesized by the HCPs that local networks and the use of eHealth applications would encourage further collaboration.

HCPs perceive a barrier when a patient needed to be referred to a non-participating practice or HCP. For these treatments, patients may be less well served as practitioners outside NPRL would not have such a detailed insight into the treatment protocols. Patients might get more biomedically oriented treatments, leading to confusion. HCPs in secondary and tertiary care thought that NPRL would especially have advantages for primary care since interdisciplinary teamwork with a focus on CMP patients is already regular care in secondary and tertiary organizations.

**Education days and practice meetings.** At the end of Phase 1, HCPs found the education days somewhat confusing. Using their feedback during these education days, the taught treatment protocol was further improved and made flexible, but it seemed that HCPs preferred a more defined protocol. Therefore, in later phases, the project team composed a more fixed treatment protocol, which was found to be clearer. Overall, they instituted a clearer layout of the education days. HCPs indicated that the visits of the project team to the primary care practices gave added value. They changed mindsets and encouraged active participation. However, after the project team left, it was difficult to maintain focus on NPRL in daily practice.

## Inner setting

**Mission and vision.** According to some HCPs, most Dutch health care practices have a more biomedical oriented vision which clashes with the biopsychosocial vision of NPRL. This may be caused by the biomedical education which they had received, as described in paragraph *Dutch culture*, *laws*, *and regulations* (see below). For this reason, some HCPs may feel misunderstood by their colleagues in their CMP treatment approach.

**Local laws and regulations.** Due to personnel shortages (for example MHPNs) and the increased workload associated with transition from secondary to primary care, HCPs in primary care have a full schedule. This hinders recruitment and active participation. In the future, the organization of care will shift towards the enlargement of primary care practices with more HCPs for the same number of patients, which could be an advantage for implementing NPRL. *"Our practice is large enough to divide projects among staff, resulting in enough time and funding to participate. I think the reorganization of general practice care towards practice enlargement will be important. With more GPs in one practice, you have time for multidisciplinary collaboration" (P10, GP, FG1).*

Additionally, current daily general practice care is unsuitable for networking on a large scale. There is a growing number of GPs with specializations but patients are connected to a practice based on geographical location, not on specialization, and often they are connected to only one GP in a practice. Primary HCPs do not often refer their patients to colleague GPs based on their specializations. Some HCPs in primary care commented on the complexity of NPRL. They said it was hard to implement all the new desired elements and protocols at once, finding it difficult to learn different tools at the same time when the general workload was also heavy.

**Collaboration with local partners.** Multidisciplinary care is not feasible for small practices in primary care because of restrictions in financing, according to the HCPs employed: i.e. their financial buffer is smaller. Some GPs have a preference for a specific therapist practice in their local network. Moreover, HCPs experience competition between physiotherapy practices and commercial rehabilitation treatment centres. As a result, practice owners neglect the screening of patients with a specific level of complexity on the assumption that this would negatively influence the number of patients able to be treated. *"I have a patient who can be treated better elsewhere, but I do not work there. I think it is good if you can neglect that, I can do that, but I am not the director who is responsible for the finance. But I think this will be a barrier for the future" (P15, RP, FG5).*

## Outer setting

**Health care insurance.** Health insurance policies in the Netherlands restrict the number of physiotherapy consultations that they will reimburse. HCPs and patients saw this as a pitfall for implementing NPRL as the consultations paid for are often insufficient to learn and apply the new self-management principles. In Dutch health care in 2018, patients may purchase additional insurance packages to cover extra physiotherapy sessions. Several different packages for different numbers of therapy sessions are available but HCPs are aware that patients with a low socioeconomic status cannot afford these. Unfortunately, the highest prevalence of CMP is amongst those patients. This affects the motivation of HCPs as well when it is already known at the beginning of treatment that the number of available consultations is insufficient.

**Reimbursing health care practices and organizations.** Multidisciplinary patient-related meetings between HCPs in primary care are not financially covered, which is a barrier for implementation. Financing and attending multidisciplinary meetings regularly is an especial problem for small practices with only a few staff members. Besides, when practices participate in more networks for various diseases, all with additional multidisciplinary meetings, this results in even heavier workloads and burdens for a primary care practice. As patients with CMP are often confronted with comorbidities, HCPs are required to attend several meetings for the same patient, making treatment and collaboration challenging.

MHPNs have an important role in NPRL as they can reduce burdens on GP. However, GPs point out that they receive little funding for deployment of a MHPN, which is not enough to cover all CMP patients who need their help. RPs, GPs, and therapists advocate future bundled payments to facilitate multidisciplinary meetings. *"I think, there should be bundled payments which also cover multidisciplinary meetings. These meetings are often with a limited number of PTs and GPs, while meetings with more disciplines and structure are needed. I think if you do not structure it with bundled payments, due to the bustle of the day, NPRL will not be rolled out more broadly." (P4, PT, FG1).*

**Dutch culture, laws, and regulations.** HCPs indicated that diagnosing someone with CMP makes the patient feel they are not being taken seriously. As CMP is an abstract phenomenon with large inter-individual variations in perception, patients often feel they are not

understood by their HCPs, family, and friends. *"Patients perceive difficulties with the fact that they are diagnosed with fibromyalgia* [a subgroup of CMP]. *When you bring this message, they are staring at you*: *they think that something is wrong with them" (P12, GP, FG2).*

Overall, current health care is biomedically oriented and HCPs not participating in NPRL often share this orientation. This makes it challenging for professionals working to NPRL guidelines to discuss the patient from a biopsychosocial viewpoint. *"I have problems with the fact that the practice I work in has a more hands-on view of treatment. . . It is difficult to convince my colleagues* [of the need] *for CMP rehabilitation" (P5, PT, FG3).* Also, CMP is not recognized as a disease in itself, causing a lack of clarity in defining which kind of care suits these patients. During HCPs' education, little attention is paid to the biopsychosocial model and/or patients with unexplained complaints. In addition, the content and amount of information varies per discipline. HCPs still have to check for red flags which indicate an underlying medical disease needing further treatment. This necessary biomedical screening is an important part of a proper biopsychosocial approach, but HCPs often see this as different to biopsychosocial screening. RPs felt that there were large number of unjustified referrals from primary care, indicating a lack of knowledge of CMP among GPs.

HCPs stated that they were more willing to participate in NPRL if the workload was not too heavy, as there was a pleasant ambiance in the collaboration with colleagues.

Additionally, frequently mentioned laws and regulations which hinder the implementation of eHealth include the new general data protection regulations (GDPR) and the inability to link ICT-systems, as these hinder data transferal.

## Individual characteristics

**Knowledge and beliefs.** Matched care is perceived as an added value by HCPs. Due to stigmatization and large variations in complexity between patients, HCPs in primary care may see patients with CMP as difficult to guide. Even after participation in the educational meetings, they wanted more training to increase their competencies to refer and treat these patients adequately. *"Maybe, more training about CMP education is necessary, so that we receive more tools to increase certainty" (P1, PT, FG1).*

In Phase 1, the HCPs in primary care reported difficulties in recognizing and quantifying the level of complexity of patients with CMP. They estimated that they only recognized 10–20% of the CMP population during consultations, as they tended to have a prototype patient with CMP in mind. *"Personally, I was frantically searching for the ideal patient to include him, following the protocol" (P3, PT, FG1).* They felt uncertain and afraid to make a false diagnosis of someone suspected to have CMP as they did not want to burden the patient unnecessarily. The fact that the group of patients in primary care is diverse with a wide variety of complaints makes recognition of CMP more difficult. In Phase 3, after additional training, HCPs found recognition easier but they still desired more experience. Also, some HCPs thought that not all patients were eager to participate in a study with questionnaires and/or eHealth and for this reason they did not invite all patients to participate.

**Background experiences.** The difference in the level of knowledge about CMP at the start of NPRL made it difficult to adjust the content and duration of training to everyone's needs. Some HCPs had prior experience with projects addressing CMP and with collaboration in primary care. This could have facilitated the implementation as they already had a more biopsychosocial orientation and collaborative experience but they were disappointed that the results of these previous projects had not been integrated into daily care processes.

**Motivation.** Reasons for HCPs to participate included providing evidence-based health care, keeping health care affordable, increasing their personal network by multidisciplinary

collaboration in a matched care setting, earlier involvement in projects for patients with CMP, the scientific basis of NPRL, or the fact that their practice owners agreed to participate. HCPs saw challenges in motivating patients to participate in a biopsychosocial treatment as, in general, patients had a more biomedical focus. For example, in physiotherapy, therapists indicated that patients expected a biomedical therapy such as massage. This led to rather low participation rates. However, some patients in the final focus group emphasized the added value of exercises. *"I really like my physiotherapist because I get a few exercises, such as riding the bike, walking, and exercises with a machine. That is going well. Afterwards I get also a massage, also really helpful"* (SP34, patient, FG6). HCPs stated that patients already receiving biomedical treatment, often for years, are less open to a change of approach. Therapists thought that, with some patients, starting treatment with a biopsychosocial approach decreased their credibility, which made them reluctant to invite them for participation. Moreover, not all patients want to be referred to secondary or tertiary care, although this might better suit the complexity of their pain complaints, because of their good relationship with their primary care therapist.

A facilitator for recruitment is an enthusiastic HCP, which makes it easier to motivate patients to participate. Conversely, when patients are eager to participate in the biopsychosocial treatment and research study, it enthuses the HCPs. *"My therapist let me see the connection between being more physically active after practising, despite the pain. When I saw this link, that was nice to see"* (P34, patient, FG6).

### Process

**Development.**   According to HCPs, the iterative, bottom-up implementation strategy suits those in primary care working in CMP as it allows adjustments to situations in daily practice. *"Most innovations use window dressing, first a lot of participating organizations, and after that development of the content. In NPRL, it looks like the other way around. First, the content development in a small network, which fits better with daily care"* (P18: PA, FG4).

An advisory board before the start of the project and the recruitment methods of HCPs were seen as facilitators. HCPs were attracted to participate in NPRL by the project group, other participating HCPs, practice owners, a local physiotherapist network, or an advertisement. HCPs found it important that a tertiary rehabilitation centre, which has expertise in pain rehabilitation, was the intervention source of NPRL. Also, multidisciplinary meetings with the project team were seen as facilitators as they changed HCPs' mindsets and reminded them of the active participation aspects. However, the subject of the meetings was often about getting started with NPRL, instead of experiences of working in NPRL. According to the HCPs, the project team used their input, had a fixed protocol, and communicated well. During the recruitment of health care practices, two local networks declined participation due to lack of time in their practice. They stated that they were too busy to implement a new project adequately.

**Implementation.**   Only three local networks participated in this study, which was however perceived as positive because, in a pilot study for complex interventions, a small group of HCPs is recommended. However, the small number of networks was also a barrier as it was difficult to collaborate and refer patients efficiently. Therefore, a critical mass of health care organizations is needed for proper implementation. Non-participating practices, organizations or colleagues lacked the multidisciplinary collaboration and shared biopsychosocial vision. For example, therapists found difficulties in the collaboration when patients, entering their practice by direct access, had to be referred for additional diagnostics to a non-participating GP.

**Transferability.**   HCPs believe that NPRL is a solution for the current gap in care for patients with CMP and they have the confidence that NPRL will be embedded in daily care.

However, at the end of Phase 3, they still felt as if they were in separate practices instead of part of a local network. *"Currently, it is not a common work method" (P12, GP, FG5).* According to the participating HCPs, in further implementation of NPRL, it will be challenging to attract HCPs with less interest in a biopsychosocial view. Nevertheless, they were willing to assist in the recruitment of new HCPs from their network of colleagues when NPRL is expanded. They also indicated that, as the organization of primary care in general shifts towards practice enlargement with more HCPs for the same amount of patients, this could be an advantage for NPRL.

## Summary

As most findings are related to several CFIR domains and constructs, extra analyses were performed. This resulted in four summaries pertaining to biopsychosocial treatment protocols and guidelines, stigmatization of CMP in society, organization of health care, and the bottom-up implementation strategy. These summaries and main findings, along with the CFIR domains and constructs, are presented in Table 2. An extensive overview can be found in S1 Table.

## Discussion

The aim of this feasibility study was to provide insight into barriers and facilitators for the development, implementation, and transferability of NPRL, and to provide insight into its perceived value and acceptability. Intervention characteristics and implementation processes appeared to have a major positive impact on NPRL implementation. The treatment protocols and guidelines within NPRL provide consistency and transparency in the collaboration as they guide a common biopsychosocial language and consensus in treatment duration, intensity, and content (see Summary 1, Table 2). Earlier studies found that successful implementation of new knowledge takes place at the individual, group and organizational levels [35]. This requires complex changes in clinical routines, collaboration among disciplines, and changes in the organization of care, or even in cultural beliefs and attitudes [36]. However, in the review of Holopainen et al. (2020), it was found that most biopsychosocial interventions to improve health care are focused on the individual skills of HCPs instead of on collaboration [37]. An added value of our study was the multidisciplinary transmural education days and practice meetings with all HCPs from primary, secondary, and tertiary care together, which also stimulated implementation at the group and organizational levels. One barrier encountered was the difference in preferences for fixed or flexible treatment protocols and guidelines. Other studies found that the professional autonomy of HCPs underlies these differences in preferences, which causes difficulties in implementation [38,39].

An important facilitator of the development and implementation of NPRL in normal rehabilitation care was the iterative and incremental design, based on key principles of user-centred design (see Summary 4, Table 2). This bottom-up strategy increases the focus on patients' and HCPs' needs. When primary users are incorporated into the iterative design process, this leads to greater usability and acceptance. However, the project team had limited experience with this design and a longer duration of the phases might have been valuable. Some of the challenges, such as the difference in terminologies and theoretical bases, seen in user-centred design with digital health records, can also be seen in this study, due to the very diverse group of HCP disciplines, heavy workloads, and varying levels of complexity of CMP [40,41].

A barrier to participation and implementation is the stigmatization of CMP in society (see Summary 2, Table 2). Earlier research found that 38% of patients living with CMP endorse internalized stigma, which reflects feelings of alienation, social withdrawal, and discriminatory

**Table 2. Summary and main findings assigned to CFIR domains and constructs.**

| Summary | Main findings | Domain | CFIR Construct |
|---|---|---|---|
| 1. Within NPRL, treatment protocols and guidelines provide consistency and transparency in collaboration of HCPs regarding biopsychosocial language and treatment intensity, duration, and content. However, the implementation of guidelines and protocols has different barriers in daily practice | 1A. The guidelines and protocols stimulate intensive collaboration between HCPs, such as consistency in biopsychosocial language and transparency in treatment duration, intensity, and content | Intervention characteristic | • Design Quality & Packaging<br>• Cost |
| | | Outer setting | - |
| | | Inner setting | • Networks & Communications |
| | | Characteristics of individuals | • Knowledge & Beliefs about the intervention<br>• Self-efficacy |
| | | Process | - |
| | 1B. HCPs experience tension between a fixed protocol and the freedom to adjust the protocol into daily practice. This is influenced by their professional preferences | Intervention characteristic | • Adaptability<br>• Complexity<br>• Design Quality & Packaging<br>• Cost |
| | | Outer setting | - |
| | | Inner setting | • Readiness for implementation<br>• Self-efficacy |
| | | Characteristics of individuals | • Knowledge & Beliefs about the intervention |
| | | Process | • Executing |
| | 1C. It is difficult to apply the guidelines about the eHealth application and assessment tools for satisfactory use in daily care | Intervention characteristic | • Relative advantage<br>• Trialability<br>• Complexity<br>• Design Quality & Packaging<br>• Cost |
| | | Outer setting | • Patient Needs & Resources |
| | | Inner setting | • Structural characteristics<br>• Readiness for implementation |
| | | Characteristics of individuals | • Knowledge & Beliefs about the intervention<br>• Self-efficacy |
| | | Process | • Executing |

(*Continued*)

**Table 2.** (Continued)

| Summary | Main findings | Domain | CFIR Construct |
|---|---|---|---|
| 2. Participation and implementation are hindered because of stigmatization of CMP in society. Moreover, HCPs' approaches are often more biomedically oriented than biopsychosocially. | 2A. In Dutch society, CMP is stigmatized because the pain is not visible. | Intervention characteristic | - |
| | | Outer setting | • Patient needs & Resources |
| | | Inner setting | - |
| | | Characteristics of individuals | • Knowledge & beliefs about the intervention |
| | | Process | - |
| | 2B. Because the biopsychosocial vision is less common, HCPs have difficulties with (early) recognition of patients with CMP in primary care. | Intervention characteristic | • Complexity<br>• Design Quality & Packaging |
| | | Outer setting | • Patient needs & Resources |
| | | Inner setting | • Culture<br>• Implementation climate |
| | | Characteristics of individuals | • Knowledge & beliefs about the intervention<br>• Self-efficacy<br>• Individual stage of change |
| | | Process | • Executing |
| | 2C. HCPs have difficulties motivating patients for a biopsychosocial treatment because the attitudes of both are more biomedically focused. | Intervention characteristic | • Complexity<br>• Design Quality & Packaging |
| | | Outer setting | - |
| | | Inner setting | - |
| | | Characteristics of individuals | • Self-efficacy |
| | | Process | - |

(*Continued*)

**Table 2.** (Continued)

| Summary | Main findings | Domain | CFIR Construct |
|---|---|---|---|
| 3. The current organization of health care for patients with CMP, such as the culture, structure, and financing of health care practices, complicates the implementation between and within the practices. | 3A. The culture of health care practices, such as the ambiance and attitude, determines the success of the collaboration between HCPs. | Intervention characteristic | - |
| | | Outer setting | • Cosmopolitanism<br>• External policy & incentives |
| | | Inner setting | • Structural characteristics<br>• Culture<br>• Implementation Climate |
| | | Characteristics of individuals | • Self-efficacy |
| | | Process | - |
| | 3B. The current organization of financing health care in the Netherlands hinders the implementation of NPRL. | Intervention characteristic | • Complexity<br>• Cost |
| | | Outer setting | • Patients' needs & Resources<br>• Cosmopolitanism<br>• External Policy & Incentives |
| | | Inner setting | • Structural Characteristics<br>• Network & Communications |
| | | Characteristics of individuals | - |
| | | Process | - |
| | 3C. The structure of the organization of health care practices in primary care is complex. | Intervention characteristic | • Adaptability<br>• Trialability<br>• Complexity<br>• Cost |
| | | Outer setting | • Cosmopolitanism<br>• Peer pressure<br>• External Policy & Incentives |
| | | Inner setting | • Structural Characteristics<br>• Networks & Communications<br>• Implementation Climate<br>• Readiness for Implementation |
| | | Characteristics of individuals | • Self-efficacy |
| | | Process | - |

(*Continued*)

**Table 2.** (Continued)

| Summary | Main findings | Domain | CFIR Construct |
|---|---|---|---|
| 4. The iterative, bottom-up implementation strategy fits with the HCPs in CMP. However, a critical mass of health care organizations is needed for proper implementation. | 4A. The active iterative, bottom-up development and participation of HCPs and the project team in the implementation process of NPRL is seen as an advantage. | Intervention characteristic | • Intervention source <br> • Evidence strength & Quality <br> • Relative Advantage <br> • Adaptability <br> • Design Quality & Packaging |
| | | Outer setting | • Implementation Climate |
| | | Inner setting | - |
| | | Characteristics of individuals | • Knowledge & Beliefs about the intervention <br> • Self-efficacy <br> • Individual identification with Organization |
| | | Process | • Engaging <br> • Executing |
| | 4B. A critical mass of health care organizations is necessary for properly implementing NPRL. | Intervention characteristic | • Complexity <br> • Design Quality & Packaging |
| | | Outer setting | - |
| | | Inner setting | • Structural characteristics <br> • Network & Communications <br> • Culture <br> • Implementation Climate |
| | | Characteristics of individuals | - |
| | | Process | - |
| | 4C. HCPs believe that NPRL is a solution to the current gap in care for patients with CMP. | Intervention characteristic | • Evidence strength & Quality <br> • Relative Advantage <br> • Adaptability |
| | | Outer setting | - |
| | | Inner setting | • Structural characteristics |
| | | Characteristics of individuals | • Knowledge & Beliefs about the intervention |
| | | Process | - |

experiences based on pain [42]. This may be caused by the fact that most CMP complaints are without underlying disease, which deviates from the widely held biomedical model. The review of De Ruddere et al. (2016) reports that individuals in the general population and HCPs such as physiotherapists and GPs, discount pain reports, take patients less seriously, and express doubt about the credibility of patients with nonmalignant pain [43]. HCPs' approaches during consultations were often more biomedical than biopsychosocial, before their participation in NPRL. In contrast to the increasing evidence for the biopsychosocial model of CMP, the majority of the HCPs have received a biomedical-focused training or education [44–46]. This biomedical training is likely to shape their attitudes and core beliefs toward CMP [47]. In our study, it has been shown that different views exist between GPs, therapists, or RPs about the biopsychosocial treatment of patients with CMP. Therapists with a biomedical orientation are more likely to advise patients to limit activities, and the attitudes and beliefs of GPs towards CMP are characterized by underuse of exercise referrals [48–51]. Where therapists hold strong biomedical beliefs about CMP, patients will tend to adopt these beliefs accordingly [48]. A finding of this study is that dynamic and flexible teams with young personnel make implementation easier as they are more likely to have been trained with a biopsychosocial vision, and they are more comfortable implementing treatment protocols. Due to this tension between biomedical and biopsychosocial visions, and the fact that primary care is more generic than specialized secondary and tertiary care, collaboration in a transmural network is challenging.

A second barrier is the organization of health care for patients with CMP, including the culture, structure, and financing of health care practices, which complicates implementation between and within practices (see Summary 3, Table 2). An optimal organization of outer and inner settings is important when implementing a new e-health technology [52]. However, in line with another study, several barriers are mentioned in the outer setting, such as a lack of integration with other electronic systems, and time constraints [53]. One reason for these time constraints is the heavier workload entailed in NPRL. Forty-four percent of the Dutch HCPs reported a heavy workload in 2019 [54]. And as many as 72% of Dutch general practices and primary health care centres reported their workloads to have increased in the last year. Because of these workloads, they had limited time to implement a new complex intervention, such as NPRL. If HCPs had more time per patient, requiring another way of financing health care, this would lead to fewer referrals [55]. Therefore, a different way of financing health care could lead to better implementation of NPRL. This is in line with Singer et al. (2011), who declared that most health care and payment systems are not designed to achieve integrated patient care [19]. Therefore, a case-manager in primary care is recommended to overcome barriers with time constraints and follow-up of patients, as they will have the resources for this [6].

Additionally, a critical mass of health care organizations is needed for proper implementation. The study sample seems to be representative of HCPs working in primary, secondary, and tertiary care with considerable variation in the context of where and how the HCPs practised. However, our convenience sample of three local networks only covers a small number of practices and is from one geographic area, and therefore may not be representative of other populations. One barrier could be that not all HCPs are open to multidisciplinary treatment in primary care or multidisciplinary collaboration with secondary and tertiary care. Moreover, not all HCPs are interested in specializing in treatment for patients with CMP, which makes the enlargement of NPRL with more HCPs difficult. However, a facilitator could be the increase of medical staff in primary care practices, with the same number of patients, which is expected to be the future in primary care [56].

A strength of this study is that it follows MRC guidance for complex interventions, which advises non-randomized feasibility studies [26]. Moreover, the iterative method is recommended to progressively refine the design before embarking on a full-scale evaluation. The

HCPs felt involved in the development of NPRL because adjustments based on the barriers and facilitators found were made after each iterative cycle. However, for further research, it is important to involve HCPs even more in the development of the intervention, which can be executed with Design Thinking. The review of Altman et al. (2018) indicates that Design Thinking may result in more usable, acceptable, and effective interventions, compared to traditional expert-driven methods [57]. And they describe Design Thinking as a promising approach for the development, implementation, and transferability of an intervention. In a further study with Design Thinking, it will be important to treat all participants as equal, reimburse them for time spent, and give them equal control over decision-making, which was not the case in this study.

In our study, patients are included via the eHealth application. The fact that some HCPs and patients did not know all the functions of the application resulted in suboptimal integration of NPRL in daily care and this may have limited the number of inclusions of patients. The eHealth application was a sub-intervention of NPRL. Because of the complexity of all sub-interventions, such as assessment tools, treatment protocol and focus on collaboration, implementation of the eHealth application in daily care was limited. Proper implementation in daily care is important before using eHealth as a recruiting strategy for patients. Besides, it is recommended that sub-interventions be implemented step-by-step instead of all at once to stimulate effective implementation in daily care. Sub-interventions must be developed based on the inner setting, which is more easily adapted to use in daily care than factors in the outer setting.

The CFIR is a comprehensive model for understanding implementation barriers and facilitators: it was used in this study to develop the topic lists and analyse the qualitative data [27]. Because of the complex interactions in the implementation of NPRL, there was an overlap in the use of domains and constructs for some results. In particular, the differences between inner and outer settings were sometimes difficult to distinguish. An example of this is the size, geographical location, and attitude towards CMP of GP and therapist practices in primary care. On the one hand, this is determined by the inner setting, the way practices develop their business plans. On the other hand, the government and authorities in the outer setting can create laws and regulations to steer these business plans. Therefore, data were assigned to all domains and constructs which were related to that part of data but summarized in the domain which reflected the best that theme. In our experience, the CFIR must be used as a flexible framework, in line with the findings of another recent study [58].

It can be concluded that NPRL seems feasible if the identified barriers and facilitators are anticipated. This study contributes to understanding factors that influence the development, implementation, and transferability of NPRL. Currently, international and national guidelines mention network collaborations as the future of care in (pain) rehabilitation, which is in line with NPRL [21,59,60]. Our implementation strategies, as well as barriers and facilitators, may assist managers and therapists in other clinical settings who aspire to implement NPRL using a similar model. Moreover, it forms the basis for refinements of NPRL. The barriers will be broken down as much as possible and facilitators will be used to plan a large-scale process and effect evaluation on Quadruple Aim outcomes such as health of the patients, (cost-)effectiveness, the satisfaction of patients with care, and meaning in the work of HCPs.

## Supporting information

**S1 Checklist.**
(PDF)

**S1 Table. Overview of main barrier and facilitator nodes per CFIR domain and construct.**
(PDF)

**S1 File. Protocol of NPRL.**
(PDF)

**S2 File. Assessment Tool 1: Primary care.**
(PDF)

**S3 File. Topic list focus groups and interviews.**
(PDF)

## Acknowledgments

We would like to thank Mario Geilen, physician assistant Adelante Centre of Expertise in Rehabilitation and Audiology, Hoensbroek, The Netherlands, for his contribution to the development of the treatment protocols and the education of the HCPs. We would like the patients for participating in the focus group. Moreover, we would like to thank Jasper Trietsch, general practitioner, and Pascalle Welman, physiotherapist, both at Gezondheidscentrum Terwinselen, Kerkrade, The Netherlands, for their reading and cross-checking of this article. Thanks also to Les Hearn for scientific and English mother-tongue proofreading and editing (les_hearn@yahoo.co.uk).

## Author Contributions

**Conceptualization:** Ivan P. J. Huijnen, Mariëlle E. A. L. Kroese, Dirk Ruwaard, Jeanine A. M. C. F. Verbunt.

**Formal analysis:** Cynthia Lamper, Gijs Brouwer.

**Funding acquisition:** Ivan P. J. Huijnen, Jeanine A. M. C. F. Verbunt.

**Investigation:** Cynthia Lamper, Albère J. Köke, Gijs Brouwer.

**Methodology:** Cynthia Lamper, Ivan P. J. Huijnen, Mariëlle E. A. L. Kroese, Dirk Ruwaard, Jeanine A. M. C. F. Verbunt.

**Project administration:** Cynthia Lamper, Albère J. Köke.

**Supervision:** Ivan P. J. Huijnen, Mariëlle E. A. L. Kroese, Jeanine A. M. C. F. Verbunt.

**Validation:** Ivan P. J. Huijnen.

**Visualization:** Cynthia Lamper.

**Writing – original draft:** Cynthia Lamper.

**Writing – review & editing:** Ivan P. J. Huijnen, Mariëlle E. A. L. Kroese, Albère J. Köke, Gijs Brouwer, Dirk Ruwaard, Jeanine A. M. C. F. Verbunt.

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
