## [Decision Letter · Decision Letter 0]

7 Apr 2021

PONE-D-20-32156

Exploring the feasibility of a network of organizations for pain rehabilitation: what are the lessons learned?

PLOS ONE

Dear Dr. Huijnen,

Thank you for submitting your manuscript to PLOS ONE. After careful consideration, we feel that it has merit but does not fully meet PLOS ONE’s publication criteria as it currently stands. Therefore, we invite you to submit a revised version of the manuscript that addresses the points raised during the review process.

The manuscript has been evaluated by two reviewers, and their comments are available below. You will see Reviewer 2 has commented on the timeliness of your manuscript. However, both reviewers have raised concerns and the manuscript will need significant revision before it can be considered for publication – you should anticipate that the reviewers will be re-invited to assess the revised manuscript, so please ensure that your revision is thorough. I have outlined some of the key concerns noted by the reviewers below, but you should respond to all concerns mentioned by the reviewers in your response-to-reviewers document. 

The key concerns noted by the reviewers are the need to expand definitions of the study terminology (e.g., “matched care approach” and “integrated transmural network”), additional information about the assessment tools, and clarity about the study sample and setting. Specifically, Reviewer 2 noted a discrepancy between the study sample and the sample size calculation presented in the previously published protocol. These issues impact the interpretation of the results and should be explored.

We look forward to receiving your revised manuscript.

Kind regards,

Danielle Poole

Staff Editor

PLOS ONE

Journal Requirements:

3. Please include additional information regarding the content validation survey or questionnaire used in the study to assess the HCPs  views on patients with CMP, their current referral pattern, and patient characteristics and ensure that you have provided sufficient details that others could replicate the analyses. Furthermore please provide a copy of the questionnaire as supporting information.

4. Please provide additional details regarding participant consent. In the ethics statement in the Methods and online submission information, please ensure that you have specified what type you obtained (for instance, written or verbal, and if verbal, how it was documented and witnessed). If your study included minors, state whether you obtained consent from parents or guardians. If the need for consent was waived by the ethics committee, please include this information.

5. Please remove the "CONFIDE" watermark that you currently have in the background of your manuscript pages.

7. We note that you have indicated that data from this study are available upon request. PLOS only allows data to be available upon request if there are legal or ethical restrictions on sharing data publicly. For information on unacceptable data access restrictions, please see http://journals.plos.org/plosone/s/data-availability#loc-unacceptable-data-access-restrictions.

8. We note that Figure 1 in your submission contain copyrighted images. All PLOS content is published under the Creative Commons Attribution License (CC BY 4.0), which means that the manuscript, images, and Supporting Information files will be freely available online, and any third party is permitted to access, download, copy, distribute, and use these materials in any way, even commercially, with proper attribution. For more information, see our copyright guidelines: http://journals.plos.org/plosone/s/licenses-and-copyright.

9. We noted in your submission details that a portion of your manuscript may have been presented or published elsewhere.

[Figure 1 Construction of the health care system in Network Pain Rehabilitation Limburg. Published in Lamper et al. (2019) [22]. Inclusion of this figure in the current submission does not constitute dual publication as the information in this figure is about the content of the intervention. The previous publication was in the protocol article. And logically, the intervention in this results article did not change from the version published in the protocol article.]

10. Thank you for stating the following in the Competing Interests section:

[IH, AK, and JV report grants from Health Insurance Companies CZ, VGZ and Achmea, during the conduct of the study. The other authors declare that there is no conflict of interest. ].

Reviewers' comments:

Reviewer's Responses to Questions

**Comments to the Author**

1. Is the manuscript technically sound, and do the data support the conclusions?

Reviewer #1: Partly

Reviewer #2: Yes

2. Has the statistical analysis been performed appropriately and rigorously? 

Reviewer #1: N/A

Reviewer #2: N/A

3. Have the authors made all data underlying the findings in their manuscript fully available?

Reviewer #1: Yes

Reviewer #2: No

4. Is the manuscript presented in an intelligible fashion and written in standard English?

Reviewer #1: No

Reviewer #2: Yes

5. Review Comments to the Author

Reviewer #1: Exploring the feasibility of a network of organizations for pain rehabilitation: what are the lessons learned?

Abstract: Avoid abbreviations in the abstract

Abstract, results: “One barrier is stigmatization…” Barrier for what?

Abstract, results: What are “non-participating” HCPs? Are these people who did not agree to participate in your study? If yes, how did you collect and analyzed their data?

Abstract, results: I do not understand what is the link between a HCP that uses approached more biomedical to lead to patients resist participating in NPRL? Was this an hypothesis of the study?

Abstract, results: The sentence that cultural, structural and financial aspects are barriers, this sentence needs a bit more details to explain what exactly are these barriers.

Abstract, results: “HCPs preferred the iterative, bottom-up strategy”. They preferred this as opposed to what other strategy?

Introduction

Line 68. Do we all know and do we all agree what is the right care, at the right place and the right time? This is a jargon that is often quoted by politicians, but I am not sure there is agreement about what this means. If this is the case, then, perhaps the right place to start would be to work on a document that outlines exactly what this right care, at the right place and right time means.

Page 75, can you please explain what “transmural” means? It is not a term commonly used where I am.

The use of abbreviations really make this introduction hard to read.

Line 79. What is the meaning of “integrated matched care”?

METHODS

Line 122. What is a “matched care approach” and what is “integrated transmural network”?

Line 124 starts explaining what is “matched care approach”, but it does not explain what is matched? Are the patients matched to a HCP and they stay with this same professional along all the way? Is this what the “matched” implies?

Line 125 starts explaining what “integrated” means, and I am still not clear what it is. Does this implies that al patients completed the same sorts of questionnaires and were all examined the same way? And that all these questionnaires were available to everyone in the same electronic medical records?

Line 138. What is transmural?

Line 140. Are GPs the same as family doctors in The Netherlands?

Line 141. Were nurse practitioners and pharmacists excluded from participating?

Line 142. What is a “local network”? Is this the same as a group practice where the physicians and non-physicians receive a fixed salary and are not reimbursed as a fee-for-service type of payment?

Line 157. Was there any accreditation or training for these HCPs in order to teach them on the best practices and clinical practice guidelines related to the management of patients with chronic musculoskeletal pain?

Line 157. Are there “pain clinics” in The Netherlands? Are these mostly for interventional pain procedures offered by anesthetists? Could patients have access to interventional pain medicine if they needed?

Line 200. I do not understand why the goal of this phase was “to organize care in daily practice”. I would say that this should be the goal of Phase 1.

Line 211. What do you mean by “small questionnaires”?

Line 240. Is this “Results”? If yes, there should be a subheading here?

Line 240. This is the first time that “Assessment tool 1 and tool 2” appear in the manuscript. Should they be described in the methods? Please describe in terms of what are they, who uses them, when they use them and why.

Line 250. Can a brief explanation of these treatment protocols and guidelines be provided in the methods?

Line 256. “reduced number of consultations prescribed”. I am not sure what this means. Can people prescribe consultations?

Line 271. This is the first time that Ehealth is mentioned in the manuscript. I suggest to describe this in the methods.

Line 282. What is the link between eHealth application and adherence to treatment? Do patients have access to their own medical records? I thought the patients were only entering data for questionnaires.

Line 286. Are patients asked to complete daily questionnaires about pain and pain intensity? Is it ethical to ask patients to do this? There is a strong body of evidence showing that the more time a person spend thinking about their pain, the higher the chances of this pain becoming chronic, it is like creating a permanent memory of pain in the brain.

Line 290. It is not clear how the collaboration was achieved. Were the patients seen by the different professionals individually in different occasions (multi-professional approach), or where the patients seen by all professions simultaneously (inter-professional approach)?

Line 299. “young and dynamic” How do you define that?

Line 300. What is a local network? Is it based on a geographical area? A funding model? A group of professionals connected by the same eHealth system? The same source of funding?

Line 304. What is a “more biomedically oriented treatment”? Could these be things like acupuncture, massage, modalities, injections, oral medications, topical creams?

Line 310. Please give details about what these education days offered.

Line 311. Who developed these protocols? Were they developed with input from primary care professionals?

Line 321. Mission and vision. Are these mission and vision for the feasibility study? For the network?

Line 322. “most health care practices” – does this refer to the practices included in the feasibility study? Or is this about practices in general in The Netherlands?

Line 468. This is the first time an advisory board is mentioned. Please mention in the methods that an advisory board was employed.

Reviewer #2: Dear authors: thank you for letting me to comment on your work

The topic of this manuscript is timely because it deals with a topic with potential for improvement in the treatment of patients with chronic musculoskeletal pain.

This study can help in the process of improving the organization of healthcare for patients with chronic pain and emphasizes the biopsychosocial approach in the management of patients.

The manuscript is well written and understandable. I just have to make a few small observations that may help improve the manuscript

The authors have followed COREG criteria for reporting qualitative research. I am happy to know this because this is a good quality criteria. Please provide response to the 32 item checklist. Most of the items are considered in the manuscript, but some are not

Another aspect that denotes quality and rigor is that the authors have previously published the protocol of the study they planned to do (BMJ Open. 2019; 9 (6): e025962). However, in this protocol it was intended to evaluate approximately 100 patients that would receive questionnaires regarding satisfaction with care and their health status and pain-related disability, and 10 patients for the focus group. Finally only 58 patients participated in the program, and 6 of them in the focus group. It would be interesting to have information about satisfaction of patients and disability to know the impact of the program on them. Also Please explain the reason, if any, for not having reached the planned number of patients

In general the writing style is friendly and well written. However the phrase about the inclusion criteria is confusing for me (page 6; lines 163-166) “They were excluded from the study if there was any suspicion of a medical (orthopaedic, rheumatic, or neurological) disease or (underlying) psychiatric disease that could explain the current pain complaints (e.g. rheumatism or hernia) and/or that could be treated by adequate existing therapy”. The conjunction and/or is misleading. If the conjunction "or" is used, it is implied that those patients with a medical diagnosis that explains symptoms will be excluded, regardless of whether they do not have effective treatment of their condition. If this is the case, for example an epicondylitis or a spondylolisthesis (both specific diagnoses, but in many cases without effective treatment) would be excluded. Please explain exclusion criteria

The author used Assessment tool 1 for HCP in primary care and Assessment tool 2 for secondary and tertiary care. I can not find Assessment tool 2 in supplementary material

In Assessment tool 1 for Primary Care, the flow chart for classifying patients is in Dutch. Please provide a translation

In conclusion, it is a very interesting article that I would like to see published. It is a feasibility study that can help clinicians and managers to implement an integrative network program for the management of patients with chronic pain

6. PLOS authors have the option to publish the peer review history of their article (what does this mean?). If published, this will include your full peer review and any attached files.

Reviewer #1: No

Reviewer #2: **Yes: **Julio Domenech

---

## [Author Response · Author response to Decision Letter 0]

2 Jun 2021

Danielle Poole

Staff Editor

PLOS ONE

Maastricht, the Netherlands, June 2, 2021

Manuscript title: Exploring the feasibility of a network of organizations for pain rehabilitation: what are the lessons learned? 

Manuscript ID: PONE-D-20-32156

Dear Mrs. Poole,

Thank you very much for considering our paper “Exploring the feasibility of a network of organizations for pain rehabilitation: what are the lessons learned?” for PLOS ONE and for having it reviewed by two reviewers in the field of chronic pain. We have carefully considered the journal requirements and reviewers’ comments, and made revisions to the paper as suggested. We appreciate the helpful feedback that was provided and believe that these comments were relevant and useful to improve our manuscript. The changes in the text are visible by track changes. Please find our reaction to the reviewers and summary of the adaptations below. Our reactions are indicated with symbol ‘>’.

We are not able to share the data (transcripts of the focus groups and interviews) publicly as there are ethical restrictions. Due to the small number of participations, geographical area, and information at our website (www.netwerkpijnrevalidatie.nl), the participations could be traceable, even if we anonymize the transcripts. Data requests can be sent to the Medical Ethics Committee Z, the Netherlands, METC 17-N-133.

As suggested by the editorial office we decided to remove fig1 from the manuscript and we refer to the original publication of the figure within the manuscript. This figure is published in our peer-reviewed and published protocol article in BMJ Open: [23]. Lamper C, Kroese M, Köke A, Ruwaard D, Verbunt J, Huijnen I. Developing the Network Pain Rehabilitation Limburg: a feasibility study protocol. BMJ Open. 2019;9(6):e025962. doi: 10.1136/bmjopen-2018-025962.

Additionally, based on the journal requirements, we have updated the competing interests section:

L857-860. IH, AK, and JV report grants from Health Insurance Companies CZ, VGZ and Achmea, during the conduct of the study. The other authors declare that there is no conflict of interest. This does not alter our adherence to PLOS ONE policies on sharing data and materials.

We hope you will reconsider our manuscript for publication. 

On behalf of all authors,

Yours sincerely,

Ivan Huijnen, PhD and Cynthia Lamper, MSc

Maastricht University

Faculty of Health, Medicine and Life Sciences

Department of Rehabilitation Medicine 

P.O. Box 616, 6200 MD Maastricht

The Netherlands

E-mail: ivan.huijnen@maastrichtuniversity.nl & cynthia.lamper@maastrichtuniversity.nl

 

Journal requirements and reviewer(s)' Comments to Author:

Comment 1: Please ensure that your manuscript meets PLOS ONE's style requirements, including those for file naming. 

> We have checked and applied the style requirements. 

Comment 2: PLOS requires an ORCID iD for the corresponding author in Editorial Manager on papers submitted after December 6th, 2016. Please ensure that you have an ORCID iD and that it is validated in Editorial Manager. To do this, go to ‘Update my Information’ (in the upper left-hand corner of the main menu), and click on the Fetch/Validate link next to the ORCID field. This will take you to the ORCID site and allow you to create a new iD or authenticate a pre-existing iD in Editorial Manager. Please see the following video for instructions on linking an ORCID iD to your Editorial Manager account: https://www.youtube.com/watch?v=_xcclfuvtxQ .

>We, as corresponding authors, have added our ORCID IDs.

Comment 3: Please include additional information regarding the content validation survey or questionnaire used in the study to assess the HCPs views on patients with CMP, their current referral pattern, and patient characteristics and ensure that you have provided sufficient details that others could replicate the analyses. Furthermore please provide a copy of the questionnaire as supporting information.

> We did not report the outcomes on the questionnaire in the result section of the manuscript. We only used the outcome on the self-constructed questionnaire as input for the focus group topics. We did not use it for evaluation purposes and therefore, it is not mentioned in the other parts of the article. For further clarification and avoiding ambiguities, we choose to remove this sentence in the manuscript (L191-193. are removed). 

Comment 4: Please provide additional details regarding participant consent. In the ethics statement in the Methods and online submission information, please ensure that you have specified what type you obtained (for instance, written or verbal, and if verbal, how it was documented and witnessed). If your study included minors, state whether you obtained consent from parents or guardians. If the need for consent was waived by the ethics committee, please include this information.

> We agree that the information about the informed consent procedure could be improved. Therefore, we adjusted the ethics and dissemination section: 

L117-118. Written and verbal informed consent was obtained from all patients and HCPs before the start of the interview or focus group. The verbal informed consent was recorded.

Comment 5: Please remove the "CONFIDE" watermark that you currently have in the background of your manuscript pages. 

> We have removed the watermark.

Comment 6: Your ethics statement should only appear in the Methods section of your manuscript. If your ethics statement is written in any section besides the Methods, please delete it from any other section. 

> We deleted the ethics section from p32 L860. 

Comment 7: We note that you have indicated that data from this study are available upon request. PLOS only allows data to be available upon request if there are legal or ethical restrictions on sharing data publicly. For information on unacceptable data access restrictions, please see http://journals.plos.org/plosone/s/data-availability#loc-unacceptable-data-access-restrictions. In your revised cover letter, please address the following prompts:

 We will update your Data Availability statement on your behalf to reflect the information you provide..

> We are not able to share the data (transcripts of the focus groups and interviews) publicly as there are ethical restrictions. Due to the small number of participations, geographical area, and information at our website (www.netwerkpijnrevalidatie.nl), the participations could be traceable, even if we anonymize the transcripts. Data requests can be sent to the Medical Ethics Committee Z, the Netherlands, METC 17-N-133. 

Comment 8: We note that Figure 1 in your submission contain copyrighted images. All PLOS content is published under the Creative Commons Attribution License (CC BY 4.0), which means that the manuscript, images, and Supporting Information files will be freely available online, and any third party is permitted to access, download, copy, distribute, and use these materials in any way, even commercially, with proper attribution. For more information, see our copyright guidelines: http://journals.plos.org/plosone/s/licenses-and-copyright .

We require you to either (1) present written permission from the copyright holder to publish these figures specifically under the CC BY 4.0 license, or (2) remove the figures from your submission.

> As suggested by the editorial office we decided to remove fig1 from the manuscript and we refer to the original publication of the figure within the manuscript.

23. Lamper C, Kroese M, Köke A, Ruwaard D, Verbunt J, Huijnen I. Developing the Network Pain Rehabilitation Limburg: a feasibility study protocol. BMJ Open. 2019;9(6):e025962. doi: 10.1136/bmjopen-2018-025962. 

Comment 9: We noted in your submission details that a portion of your manuscript may have been presented or published elsewhere.

As suggested by the editorial office we decided to remove fig1 from the manuscript and we refer to the original publication of the figure within the manuscript. 23. Lamper C, Kroese M, Köke A, Ruwaard D, Verbunt J, Huijnen I. Developing the Network Pain Rehabilitation Limburg: a feasibility study protocol. BMJ Open. 2019;9(6):e025962. doi: 10.1136/bmjopen-2018-025962. 

Comment 10: Thank you for stating the following in the Competing Interests section: [IH, AK, and JV report grants from Health Insurance Companies CZ, VGZ and Achmea, during the conduct of the study. The other authors declare that there is no conflict of interest. ].

Please know it is PLOS ONE policy for corresponding authors to declare, on behalf of all authors, all potential competing interests for the purposes of transparency. PLOS defines a competing interest as anything that interferes with, or could reasonably be perceived as interfering with, the full and objective presentation, peer review, editorial decision-making, or publication of research or non-research articles submitted to one of the journals. Competing interests can be financial or non-financial, professional, or personal. Competing interests can arise in relationship to an organization or another person. Please follow this link to our website for more details on competing interests: http://journals.plos.org/plosone/s/competing-interests.

> We have updated the competing interests section:

L857-860. IH, AK, and JV report grants from Health Insurance Companies CZ, VGZ and Achmea, during the conduct of the study. The other authors declare that there is no conflict of interest. This does not alter our adherence to PLOS ONE policies on sharing data and materials.

Reviewer 1: 

Comment 1: Abstract: Avoid abbreviations in the abstract. 

> We thank the reviewer for reviewing our manuscript and his/her comments. In order to increase the readability of the abstract, we have removed the abbreviations. 

Comment 2: Abstract, results: “One barrier is stigmatization…” Barrier for what?.

> We have changed the sentence to make it easier understandable.

One mentioned barrier is the stigmatization of chronic pain by the general population.

Comment 3: Abstract, results: What are “non-participating” HCPs? Are these people who did not agree to participate in your study? If yes, how did you collect and analyzed their data?.

> With non-participating HCPs we mean the HCPs working in regular care. For clarity we have changed the sentence. 

In regular care, approaches are often more biomedical than biopsychosocial, causing patients to resist participating.

Comment 4: Abstract, results: I do not understand what is the link between a HCP that uses approached more biomedical to lead to patients resist participating in NPRL? Was this an hypothesis of the study?.

> Regular care is often more biomedical oriented, searching for a physical cause of the pain. As in patients with CMP often no physical cause can be found, a biopsychosocial approach seems more suitable. However, patients do not expect a biopsychosocial approach from their HCP. Therefore, HCPs experienced barriers in convincing patients for a biopsychosocial treatment as purposed in the Network Pain Rehabilitation Limburg. This caused a lower number of patients treated in the network than expected. 

The aim of this study was to provide insight into barriers and facilitators for the development, implementation, and transferability of NPRL. The recruitment of patients was found to be a barrier for implementation and transferability of NPRL, but the resistance of patients to participate was not a hypothesis of this study.. 

Comment 5: Abstract, results: The sentence that cultural, structural and financial aspects are barriers, this sentence needs a bit more details to explain what exactly are these barriers. 

> We agree that this sentence needs more explanation. Due to the word limit of the abstract we have chosen to remove the part: with cultural, structural, and financial aspects. More information about this topic can be found in the results section of the manuscript. 

Comment 6: Abstract, results: “HCPs preferred the iterative, bottom-up strategy”. They preferred this as opposed to what other strategy? 

> To improve the clarity, we changed the sentence:

Health care professionals were enthusiastic about the iterative, bottom-up development. 

Comment 7: Line 68. Do we all know and do we all agree what is the right care, at the right place and the right time? This is a jargon that is often quoted by politicians, but I am not sure there is agreement about what this means. If this is the case, then, perhaps the right place to start would be to work on a document that outlines exactly what this right care, at the right place and right time means.

> We agree that different definitions exists for the right care, at the right place, at the right time. The Dutch National Care Standard for Chronic Pain has described this. In this study, and in the conduct of NPRL, we used this document as guideline. We have changed the sentence by adding this document as reference: 

L. 67-69. Therefore, patients with CMP often do not receive the right care, at the right place, at the right time, as described in the National Care Standard for Chronic Pain, the Netherlands [6]. 

Comment 8: Page 75, can you please explain what “transmural” means? It is not a term commonly used where I am..

> Transmural means across different level of care, for example between primary, secondary, and tertiary care. The complete definition is explained in this whole paragraph. To be consisted, we have added the word integrated in sentence 76:

The World Health Organization recommends networks in integrated transmural rehabilitation care as future developments [21].

Comment 9: The use of abbreviations really make this introduction hard to read. 

> We concur that the number of abbreviations in the introduction is high. Therefore, we have removed the abbreviations: World Health Organization (WHO) and National Care Standard for Chronic Pain (NCSCP). 

Comment 10: Line 79. What is the meaning of “integrated matched care”?

> The definition of matched care was indeed not clearly explained. Therefore, we have added this definition with a reference: 

L83-84. Matched care comprises identifying patients at higher risk. However, unlike stratified care, it tailors the intervention to the individual patient's specific existing complaints and risk factors [22, 23]. 

22. Linton SJ, Nicholas M, Shaw W. Why wait to address high-risk cases of acute low back pain? A comparison of stepped, stratified, and matched care. Pain. 2018;159(12):2437-41. Epub 2018/06/16. doi: 10.1097/j.pain.0000000000001308. PubMed PMID: 29905653.

23. Nijs J, George SZ, Clauw DJ, Fernández-de-las-Peñas C, Kosek E, Ickmans K, et al. Central sensitisation in chronic pain conditions: latest discoveries and their potential for precision medicine. The Lancet Rheumatology. 2021. doi: https://doi.org/10.1016/S2665-9913(21)00032-1.

Comment 11: Line 122. What is a “matched care approach” and what is “integrated transmural network”?

> For the definition of a matched care approach, see our response to comment 10. For the definition of an integrated transmural network, see our response to comment 8.

Comment 12: Line 124 starts explaining what is “matched care approach”, but it does not explain what is matched? Are the patients matched to a HCP and they stay with this same professional along all the way? Is this what the “matched” implies?

> We want to refer to our response at comment 10. This means that a patient is treated by the best suitable treatment for these patient, fitting his/her existing CMP complaints and risk factors. And not, as in stratified care, starting with a treatment in primary care and when not successful followed by referral succeeded to secondary or tertiary care. In a matched care approach, patients are directly referred to a HCP in tertiary care, when this matches their complaints and is thus needed. 

Comment 13: Line 125 starts explaining what “integrated” means, and I am still not clear what it is. Does this implies that all patients completed the same sorts of questionnaires and were all examined the same way? And that all these questionnaires were available to everyone in the same electronic medical records?

> We explained the term ‘integrated’ in the Introduction (L.70-77). We follow the definition of the World Health Organization: the management and delivery of health services so that clients receive a continuum of preventive and curative services, according to their needs over time and across different levels of the health system’’. This means that the different levels of health care work together (are ‘integrated’) to deliver treatment for patients with CMP. 

Comment 14: Line 138. What is transmural?

> For the explanation of transmural, we want to refer to our answer at comment 8. 

Comment 15: Line 140. Are GPs the same as family doctors in The Netherlands?

> In the Netherlands, GPs are indeed the family doctors in primary care. To increase the clarity and to make it easier comparable for other countries with different health care systems, we have added this to the sentence:

L147. For inclusion in a local network, it was necessary to have a GP (general practitioner, family doctor) participating, with in addition at least one physiotherapist (PT) or exercise therapist (ET), and, optionally, a mental health practice nurse (MHPN).

Comment 16: Line 141. Were nurse practitioners and pharmacists excluded from participating?

> At the moment, nurse practitioners and pharmacists are not involved in the rehabilitation treatment for patients with CMP in primary care in the Netherlands. It is expected that they will get involved in the future. However, as they have no prominent role in rehabilitation, we have not in- or excluded them in this study. In our discussion, we refer to the increase of medical staff in primary care practices (L.584). A nurse practitioner or pharmacists could be one of the disciplines working in primary care in the future. 

Comment 17: Line 142. What is a “local network”? Is this the same as a group practice where the physicians and non-physicians receive a fixed salary and are not reimbursed as a fee-for-service type of payment?

> We mean with a local network the HCPs in the same village or city district. (L.143. In primary care, local networks were set up in villages or city districts with local HCPs.) They exists of GPs, PTs, ETs, or MHPNs, depending on the availability of the disciplines in the area. It could be a joint practice were a GP, a PT and a MHPNs are working, but it could also be separate practices in the same area. They are non financed or reimbursed in the same way. This depends on their discipline. 

Comment 18: Line 157. Was there any accreditation or training for these HCPs in order to teach them on the best practices and clinical practice guidelines related to the management of patients with chronic musculoskeletal pain?

> We agree that the process of accreditation or training the HCPs is not described clearly. Therefore, we have added this in L155-157. Moreover, in the exclusion criteria of the healthcare professionals (L149-151) is mentioned how many education days they had to follow in order to participate. 

L159. HCPs were educated in the clinical guidelines as described in the National Care Standard for Chronic Pain and in the study process before its start and participated in focus groups at the end of each phase.

Comment 19: Line 157. Are there “pain clinics” in The Netherlands? Are these mostly for interventional pain procedures offered by anesthetists? Could patients have access to interventional pain medicine if they needed?

> In the Netherlands, ‘pain clinics’ exists and they offer different kind of treatments such as interventional pain medicine by anesthetists. As this is not the goal of Network Pain Rehabilitation Limburg, we did not include these pain clinics in our study. In our study only rehabilitation centers are included which focusses on improving daily activities and participation of patients. When patients had a preference for treatment at an anesthetists, they were referred to them outside the Network Pain Rehabilitation Limburg. 

Comment 20: Line 200. I do not understand why the goal of this phase was “to organize care in daily practice”. I would say that this should be the goal of Phase 1

> In phase 1 (development phase) the goal was to develop the way this care has to be organized. Subjects included were: which HCPs need to participate, what will be the referral patterns, which treatments will be offered. In phase 3 (transferability phase) the goal was to organize this care in daily practice so the HCPs could work with the developed NPRL of phase 1. In phase 3 the focus is on the practical aspects of participating and working in NPRL. 

Comment 21: Line 211. What do you mean by “small questionnaires”?

> We mean that the HCPs filled in a questionnaire with a few predefined questions about the treatment of each patient. These questions comprised subjects as; number of consultations, did the patient reach the treatment goal, and barriers or facilitators of the treatment. To clarify this in the manuscript, we have adjusted the sentence:

L220. After completing treatments, HCPs submitted predefined questionnaires or logbooks about the treatment (number of consultations, barriers and facilitators during treatment, achievement of treatment goal) of each individual patient.

Comment 22: Line 240. Is this “Results”? If yes, there should be a subheading here?

> Yes, this is the results section, the subheading “Results” is placed at L.236, before the table. PLOS ONE requires this type of subheadings.

Comment 23: Line 240. This is the first time that “Assessment tool 1 and tool 2” appear in the manuscript. Should they be described in the methods? Please describe in terms of what are they, who uses them, when they use them and why.

> Assessment tool 1 and assessment tool 2 are described in the “Methods” section:

L133. Two assessment tools supported the decision-making for problem and complexity mapping and treatment selection, based on the patient’s biopsychosocial profile. GPs and therapists in primary care used Assessment Tool 1 (Supplementary file 1); Assessment Tool 2 was used by RPs in secondary and tertiary care.

Comment 24: Line 250. Can a brief explanation of these treatment protocols and guidelines be provided in the methods?

> A brief explanation of the treatment protocols and guidelines is provided in the Methods section: 

L.137. In the individualized treatment plan, the patient together with the HCP set activity- and participation-related goals.

Moreover, an extended explanation can be found in our published protocol article:

23. Lamper C, Kroese M, Köke A, Ruwaard D, Verbunt J, Huijnen I. Developing the Network Pain Rehabilitation Limburg: a feasibility study protocol. BMJ Open. 2019;9(6):e025962. doi: 10.1136/bmjopen-2018-025962

Comment 25: Line 256. “reduced number of consultations prescribed”. I am not sure what this means. Can people prescribe consultations?

> We mean, the number of consultations as prescribed in the treatment protocol. To improve the clarity of the sentence, we have adjusted it:

L.267. Due to the restricted number of consultations prescribed in the treatment protocol of NPRL as compared to care as usual, some therapists in primary care indicated fear that this would lead to a drop in income from that achieved before NPRL’s start.

Comment 26: Line 271. This is the first time that Ehealth is mentioned in the manuscript. I suggest to describe this in the methods.

> We described the eHealth application in L.134-136. More information can also be found in our protocol article. 

L138. An e-health application was integrated into matched care protocols for every setting with the primary goal of supporting pain education and self-management by the patient.

23. Lamper C, Kroese M, Köke A, Ruwaard D, Verbunt J, Huijnen I. Developing the Network Pain Rehabilitation Limburg: a feasibility study protocol. BMJ Open. 2019;9(6):e025962. doi: 10.1136/bmjopen-2018-025962

Comment 27: Line 282. What is the link between eHealth application and adherence to treatment? Do patients have access to their own medical records? I thought the patients were only entering data for questionnaires.

> We mean that the eHealth application is an extra element of the treatment. It supports the treatments in primary care as it provides pain education and self-management exercises for patients (L134-136; see also comment 26). We made adjustments to clarify this sentence:

L293. All patients agreed that the eHealth application (existing of pain education and self-management exercises) stimulated them to adhere to the treatment.

Comment 28: Line 286. Are patients asked to complete daily questionnaires about pain and pain intensity? Is it ethical to ask patients to do this? There is a strong body of evidence showing that the more time a person spend thinking about their pain, the higher the chances of this pain becoming chronic, it is like creating a permanent memory of pain in the brain.

> The diaries in the eHealth application comprised questions about their activity level and participation during that day, which are rehabilitation goals. Only, once a week a question was asked about the pain intensity. We do not think that asking pain intensity in this frequency influences the chance of pain becoming chronic.

Comment 29: Line 290. It is not clear how the collaboration was achieved. Were the patients seen by the different professionals individually in different occasions (multi-professional approach), or where the patients seen by all professions simultaneously (inter-professional approach)? 

> Patients were seen interdisciplinary, simultaneously with the same treatment goal and interim consultation, by different HCPs in the local networks. For clarification, we adjusted some sentences:

L302. Some therapists appreciated the fact that interdisciplinary collaboration with GPs and MHPNs working closely together is a major pillar of NPRL

L306. Better interdisciplinary collaboration was achieved in local networks based in one site, compared to those in which the GP and MHPN were located at a different address from the therapists.

Comment 30: Line 299. “young and dynamic” How do you define that?

> The HCPs mentioned that a team with young and dynamic staff members facilitates implementation, as they are often more flexible in adapting their treatment strategy and are more willing to interdisciplinary collaboration. This is how the HCPs perceived it themselves. 

Comment 31: Line 300. What is a local network? Is it based on a geographical area? A funding model? A group of professionals connected by the same eHealth system? The same source of funding?

> See our response to comment 17.

Comment 32: Line 304. What is a “more biomedically oriented treatment”? Could these be things like acupuncture, massage, modalities, injections, oral medications, topical creams

> With a “more biomedically oriented treatment” we indeed mean one of the treatments you mention. While in a biopsychosocial treatment also attention is paid to the psychosocial status and possible complaints of the patient. 

Comment 33: Line 310. Please give details about what these education days offered.

> The content of the education days is mentioned in L155 of the Methods section and in our protocol article: 

L159. HCPs were educated in the clinical guidelines as described in the National Care Standard for Chronic Pain and in the study process before its start and participated in focus groups at the end of each phase.

23. Lamper C, Kroese M, Köke A, Ruwaard D, Verbunt J, Huijnen I. Developing the Network Pain Rehabilitation Limburg: a feasibility study protocol. BMJ Open. 2019;9(6):e025962. doi: 10.1136/bmjopen-2018-025962

Comment 34: Line 311. Who developed these protocols? Were they developed with input from primary care professionals?

> The protocols were developed by the project team with input from the advisory board, based on the National Care Standard for Chronic Pain before the start of the study. The project team consisted of researchers and health care professionals with prior knowledge of the development of treatment protocols for patients with CMP. The advisory board consisted of working in primary, secondary, as well as tertiary care. During Phase 1, these protocols were adjusted based on practical usage by the HCPs. 

Comment 35: Line 322. “most health care practices” – does this refer to the practices included in the feasibility study? Or is this about practices in general in The Netherlands?

> We mean the health care practices in the Netherlands. Therefore, we adjusted L329:

L333. According to some HCPs, most Dutch health care practices have a more biomedical oriented vision which clashes with the biopsychosocial vision of NPRL

Comment 36: Line 468. This is the first time an advisory board is mentioned. Please mention in the methods that an advisory board was employed.

> We agree that this is not mentioned in the Methods section, therefore we have adjusted L124:

L125: NPRL is described more extensively in Lamper et al. (2019): it was developed by a project team consisting of the authors CL, IH, AK, JV and an advisory board consisting of interested HCPs. The project team as well as advisory board consisted of HCPs of different disciplines and they had experience in the development of treatment protocols. 

Reviewer 2: 

Comment 1: Dear authors: thank you for letting me to comment on your work. The topic of this manuscript is timely because it deals with a topic with potential for improvement in the treatment of patients with chronic musculoskeletal pain. This study can help in the process of improving the organization of healthcare for patients with chronic pain and emphasizes the biopsychosocial approach in the management of patients. The manuscript is well written and understandable. I just have to make a few small observations that may help improve the manuscript. 

> We thank Mr. Julio Domenech for reviewing our manuscript and his compliments on our manuscript.

Comment 2: The authors have followed COREG criteria for reporting qualitative research. I am happy to know this because this is a good quality criteria. Please provide response to the 32 item checklist. Most of the items are considered in the manuscript, but some are not.

> We did not provide information about the repeat interviews (No 18). In this iterative design we conducted several focus groups with the same HCPs, and at least two focus groups at the same time. Therefore, repeat interviews were not necessary to conduct. Moreover, we did not provide information about the return of transcripts to the participants (No 23), as we did not consider this. We did perform a member-check with two HCPs of our manuscript (L. 233). They read the results section and checked if it reflected the discussions during the focus groups. Additionally, in our case, No 22 about data saturation was not applicable, as we interviewed all HCPs participating in NPRL at least once. We assumed that, due to the variation in HCPs participating in NPRL, all different views were discussed. As we interviewed all HCPs, it was not possible to recruit more HCPs in case data-saturation was not reached. 

Comment 3: Another aspect that denotes quality and rigor is that the authors have previously published the protocol of the study they planned to do (BMJ Open. 2019; 9 (6): e025962). However, in this protocol it was intended to evaluate approximately 100 patients that would receive questionnaires regarding satisfaction with care and their health status and pain-related disability, and 10 patients for the focus group. Finally only 58 patients participated in the program, and 6 of them in the focus group. It would be interesting to have information about satisfaction of patients and disability to know the impact of the program on them. Also Please explain the reason, if any, for not having reached the planned number of patients. 

> It has to be taken into account that, in our protocol article, a study was described to get insight into the barriers and facilitators, perceived value, acceptability and implementation strategies for the development, implementation and transferability of the NPRL. In this manuscript we reflect only on the barriers and facilitators of NPRL, which is a part of the total study as described in the protocol article. Therefore, in this manuscript, no data on their health status and pain-related disability is published. Furthermore, questionnaire data on satisfaction with care was only available of a small number of patients. As this is not representative for the patient sample presented in this study, we choose not to include these results in this article. The eHealth application was not good integrated in daily practice, and the questionnaires were collected via this eHealth application, which could be the reason of this loss-of-follow-up data. In our discussion we reflect on the fact that only a small number of patients participated in the study of NPRL: 

L607. In our study, patients are included via the eHealth application. The fact that some HCPs and patients did not know all the functions of the application resulted in suboptimal integration of NPRL in daily care and this may have limited the number of patients included.

We expected to ask approximately 10 patients for the focus group of this study. We found eight patients who agreed for attendance. However, on the day of the focus group one patient was ill, and one patient did forget the appointment. Due to the short time available, it was not possible to invite more patients. After evaluation of the data we found that six patients was enough to reach data-saturation. Therefore, we did not execute an additional focus group. 

Comment 4: In general the writing style is friendly and well written. However the phrase about the inclusion criteria is confusing for me (page 6; lines 163-166) “They were excluded from the study if there was any suspicion of a medical (orthopaedic, rheumatic, or neurological) disease or (underlying) psychiatric disease that could explain the current pain complaints (e.g. rheumatism or hernia) and/or that could be treated by adequate existing therapy”. The conjunction and/or is misleading. If the conjunction "or" is used, it is implied that those patients with a medical diagnosis that explains symptoms will be excluded, regardless of whether they do not have effective treatment of their condition. If this is the case, for example an epicondylitis or a spondylolisthesis (both specific diagnoses, but in many cases without effective treatment) would be excluded. Please explain exclusion criteria. 

> We agree that this sentence needs clarification. Therefore, we rephrased the sentence: 

L 172. They were excluded from the study if there was any suspicion of a biomedical (orthopaedic, rheumatic, or neurological) disease that could explain the current pain complaints and could be treated by adequate existing therapy. In addition, they were excluded if there was any (underlying) psychiatric disease (Personality disorder, schizophrenia, or clinical depression) that limit the possibility for behavioral change.

Comment 5: The author used Assessment tool 1 for HCP in primary care and Assessment tool 2 for secondary and tertiary care. I can not find Assessment tool 2 in supplementary material. 

> Assessment tool 2 is not added to the supplementary materials. This assessment tool was developed from our group and, in the future, we would like to publish this tool in an upcoming publication. Therefore, we are not able to share or publish the content of this tool at the moment. 

Comment 6: In Assessment tool 1 for Primary Care, the flow chart for classifying patients is in Dutch. Please provide a translation. 

> An updated, translated, version of the assessment tool 1 is uploaded as supplementary material. 

Comment 7: In conclusion, it is a very interesting article that I would like to see published. It is a feasibility study that can help clinicians and managers to implement an integrative network program for the management of patients with chronic pain

> We would like to thank you for your comment.

---

## [Decision Letter · Decision Letter 1]

4 May 2022

PONE-D-20-32156R1Exploring the feasibility of a network of organizations for pain rehabilitation: what are the lessons learned?PLOS ONE

Dear Dr. Lamper,

Thank you for submitting your manuscript to PLOS ONE. After careful consideration, we feel that it has merit but does not fully meet PLOS ONE’s publication criteria as it currently stands. Therefore, we invite you to submit a revised version of the manuscript that addresses the points raised during the review process.

We look forward to receiving your revised manuscript.

Kind regards,

Jianhong Zhou

Staff Editor

PLOS ONE

Journal Requirements:

Reviewers' comments:

Reviewer's Responses to Questions

**Comments to the Author**

1. If the authors have adequately addressed your comments raised in a previous round of review and you feel that this manuscript is now acceptable for publication, you may indicate that here to bypass the “Comments to the Author” section, enter your conflict of interest statement in the “Confidential to Editor” section, and submit your "Accept" recommendation.

Reviewer #3: (No Response)

2. Is the manuscript technically sound, and do the data support the conclusions?

Reviewer #3: Yes

3. Has the statistical analysis been performed appropriately and rigorously? 

Reviewer #3: Yes

4. Have the authors made all data underlying the findings in their manuscript fully available?

Reviewer #3: Yes

5. Is the manuscript presented in an intelligible fashion and written in standard English?

Reviewer #3: Yes

6. Review Comments to the Author

Reviewer #3: The study was extensively revised. However, to make it more understandable in my opinion it could be useful to add as supporting information what is written in the published protocol: in particular, the explanation of the primary, secondary and tertiary care with the short related protocol.

7. PLOS authors have the option to publish the peer review history of their article (what does this mean?). If published, this will include your full peer review and any attached files.

Reviewer #3: No

---

## [Author Response · Author response to Decision Letter 1]

2 Jun 2022

Journal requirements and reviewer(s)' Comments to Author:

Comment 1: Please review your reference list to ensure that it is complete and correct. If you have cited papers that have been retracted, please include the rationale for doing so in the manuscript text, or remove these references and replace them with relevant current references. Any changes to the reference list should be mentioned in the rebuttal letter that accompanies your revised manuscript. If you need to cite a retracted article, indicate the article’s retracted status in the References list and also include a citation and full reference for the retraction notice.

> We have checked all references and they are all still existing. 

Reviewer 3: 

Comment 1: The study was extensively revised. However, to make it more understandable in my opinion it could be useful to add as supporting information what is written in the published protocol: in particular, the explanation of the primary, secondary and tertiary care with the short related protocol.

> We thank reviewer 3 for this comment. We agree that an explanation of the healthcare lines and content of NPRL is of added value for the understandability of our study. Therefore, we have added a summary of our published protocol in Supplementary file 1.We adjusted L124 with a referral to the Supporting file. 

L124. NPRL is described more extensively in Lamper et al. (2019) (a summery can be found in Supporting file 1): it was developed by a project team consisting of the authors CL, IH, AK, JV and an advisory board consisting of interested HCPs

---

## [Decision Letter · Decision Letter 2]

13 Jul 2022

PONE-D-20-32156R2Exploring the feasibility of a network of organizations for pain rehabilitation: what are the lessons learned?PLOS ONE

Dear Dr. Lamper,

Thank you for submitting your manuscript to PLOS ONE. After careful consideration, we feel that it has merit but does not fully meet PLOS ONE’s publication criteria as it currently stands. Therefore, we invite you to submit a revised version of the manuscript that addresses the points raised during the review process.

Specifically, the in-house editorial staff feels that your study should be registered as a clinical trial. In particular, the participants were subject to changes in treatment approach, and health related outcomes were collected as part of the study. There is no need to provide clinical trial related documents (e.g. trial protocol or the CONSORT flow diagram) and to report the work as a clinical trial (i.e. no need to change article type into "clinical trial"), but we do request having your study registered before we proceed. Please see the WHO list of approved registries at http://www.who.int/ictrp/network/primary/en/index.html and more information on trial registration at http://www.icmje.org/about-icmje/faqs/clinical-trials-registration/). Please state the name of the registry and the registration number (e.g. ISRCTN or ClinicalTrials.gov) in the

submission data and on the title page of your manuscript when you resubmit. If you do not feel your study should be registered, please contact the journal office (plosone@plos.org) and explain your reasons.

We look forward to receiving your revised manuscript.

Kind regards,

Jianhong Zhou

Staff Editor

PLOS ONE

Reviewers' comments:

Reviewer's Responses to Questions

**Comments to the Author**

1. If the authors have adequately addressed your comments raised in a previous round of review and you feel that this manuscript is now acceptable for publication, you may indicate that here to bypass the “Comments to the Author” section, enter your conflict of interest statement in the “Confidential to Editor” section, and submit your "Accept" recommendation.

Reviewer #3: All comments have been addressed

2. Is the manuscript technically sound, and do the data support the conclusions?

Reviewer #3: Yes

3. Has the statistical analysis been performed appropriately and rigorously? 

Reviewer #3: N/A

4. Have the authors made all data underlying the findings in their manuscript fully available?

Reviewer #3: Yes

5. Is the manuscript presented in an intelligible fashion and written in standard English?

Reviewer #3: Yes

6. Review Comments to the Author

Reviewer #3: The revised form of this study has been extensively modified. In this way in my opinion the study can be published

7. PLOS authors have the option to publish the peer review history of their article (what does this mean?). If published, this will include your full peer review and any attached files.

Reviewer #3: No

---

## [Author Response · Author response to Decision Letter 2]

21 Jul 2022

Jianhong Zhou

Staff Editor

PLOS ONE

Maastricht, the Netherlands, July 21, 2022

Manuscript title: Exploring the feasibility of a network of organizations for pain rehabilitation: what are the lessons learned?

Manuscript ID: PONE-D-20-32156

Dear Jianhong Zhou,

Thank you very much for considering our paper “Exploring the feasibility of a network of organizations for pain rehabilitation: what are the lessons learned?” for PLOS ONE and for having it reviewed a third reviewer in the field of chronic pain. Please find our reaction to the reviewer and summary of the adaptations below. Our reactions are indicated with symbol ‘>’.

We have added the study registration data on the title page of the manuscript. The study is registered at the Netherlands Trial Register at 18/08/2017 (Registration number: NTR6654 or https://trialsearch.who.int/Trial2.aspx?TrialID=NTR6654).

We hope we have met all requirements for publication of the manuscript and we hope to receive a positive response to our re-submission soon.

On behalf of all authors,

Yours sincerely,

Cynthia Lamper, MSc

Maastricht University

Faculty of Health, Medicine and Life Sciences

Department of Rehabilitation Medicine

P.O. Box 616, 6200 MD Maastricht

The Netherlands

E-mail: cynthia.lamper@maastrichtuniversity.nl

Journal requirements and reviewer(s)' Comments to Author:

Comment 1: Specifically, the in-house editorial staff feels that your study should be registered as a clinical trial. In particular, the participants were subject to changes in treatment approach, and health related outcomes were collected as part of the study. There is no need to provide clinical trial related documents (e.g. trial protocol or the CONSORT flow diagram) and to report the work as a clinical trial (i.e. no need to change article type into "clinical trial"), but we do request having your study registered before we proceed.

> We have added the study registration data on the title page of the manuscript. The study is registered at the Netherlands Trial Register at 18/08/2017 (Registration number: NTR6654 or https://trialsearch.who.int/Trial2.aspx?TrialID=NTR6654).

Reviewer 3:

Comment 1: The revised form of this study has been extensively modified. In this way in my opinion the study can be published

> We thank reviewer 3 for this comment

---

## [Editor Report · Decision Letter 3]

2 Aug 2022

Exploring the feasibility of a network of organizations for pain rehabilitation: what are the lessons learned?

PONE-D-20-32156R3

Dear Dr. Lamper,

We’re pleased to inform you that your manuscript has been judged scientifically suitable for publication and will be formally accepted for publication once it meets all outstanding technical requirements.

Kind regards,

Jianhong Zhou

Staff Editor

PLOS ONE
---

## [Editor Report · Acceptance letter]

26 Aug 2022

PONE-D-20-32156R3 

Exploring the feasibility of a network of organizations for pain rehabilitation: what are the lessons learned? 

Dear Dr. Lamper:

I'm pleased to inform you that your manuscript has been deemed suitable for publication in PLOS ONE. Congratulations! Your manuscript is now with our production department. 

Kind regards, 

on behalf of

Jianhong Zhou 

Staff Editor

PLOS ONE